

# Nutrient composition of *Chenopodium formosanum* Koidz. bran: Fractionation and bioactivity of its soluble active polysaccharides

Yaw-Bee Ker[1], Hui-Ling Wu[1], Kuan-Chou Chen[2,3,4] and Robert Y. Peng[2]

[1] Department of Food and Applied Technology, Hungkuang University, Taichung City, Taiwan
[2] Graduate Institute of Clinical Medicine, College of Medicine, Taipei Medical University, Taipei, Taiwan
[3] Department of Urology, Taipei Medical University Shuang-Ho Hospital, Taipei, Taiwan
[4] TMU-Research Center of Urology and Kidney, Taipei Medical University, Taipei, Taiwan

## ABSTRACT

**Background**. *Chenopodium formosanum* Koidz. Amaranthaceae—also known as Djulis or red quinoa (RQ)—is a cereal plant indigenous to Taiwan, known for its high nutrient value. However, its bran is considered a waste product and the nutrient value has never been analyzed.

**Methods**. In this study, we examined the proximate composition of RQ bran, specifically its soluble polysaccharide fractions.

**Results**. RQ bran exhibited high contents of protein (16.56%), ash (7.10%), carbohydrate (60.45%), total polyphenolics (1.85%), betaxanthin (9.91 mg/100 g of RQ bran), and indicaxanthin (7.27 mg/100 g of RQ bran). Specifically, it was rich in polyunsaturated fatty acids (PUFAs; 39.24%)—with an n-6/n-3 and PUFA/saturated fatty acid (SFA) ratio of 18.137 and 0.743, respectively. Four soluble polysaccharide fractions were also obtained: CF-1, CF-2, CF-3, and CF-4, with yields of 3.90%, 6.74%, 22.28%, and 0.06%, respectively, and molecular weights of 32.54, 24.93, 72.39, and 55.45 kDa, respectively. CF-1, CF-2, CF-3, and CF-4 had respectively 15.67%, 42.41%, 5.44%, and 14.52% peptide moiety content and 38.92%, 50.70%, 93.76%, and 19.80% carbohydrate moiety. In CF-2, the glucose content was 95.86 mol% and that of leucine was 16.23%, implicating the presence of a typical leucinoglucan. All four polysaccharide fractions lacked glutamic acid and hydroxyproline. The IC50 of CF-1, CF-2, and CF-3 was respectively 12.05, 3.98, and 14.5 mg/mL for DPPH free radical–scavenging ability; 5.77, 4.10, and 7.03 mg/mL for hydrogen peroxide–scavenging capability; 0.26, 0.05, and 0.19 mg/mL for $O_2-$ free radical–scavenging capability; and 100.41, 28.12, and 29.73 mg/mL for Fe2+ chelation.

**Conclusion**. Our results indicated that RQ bran has a large amount of nutrient compounds, and a cost-efficient process for their extraction is needed. Their biomedical application as nutraceuticals also warrants further investigation.

Corresponding author
Kuan-Chou Chen,
kuanchou@s.tmu.edu.tw

## INTRODUCTION

Quinoa is a pseudocereal and is botanically related to spinach and amaranth (*Amaranthus* spp.) (*Fuentes et al., 2008*). Originating in the Andean region of northwestern South America, quinoa was domesticated 3,000 to 4,000 years ago for human consumption. Archaeological evidence demonstrates quinoa was used for livestock feed 5,200–7,000 years ago (*Jancurvá, Minarovičová & Dandár, 2009*). In 1940, red quinoa (RQ)—also known by its indigenous name, Djulis—was verified by Japanese researchers to be a quinoa variety native to Taiwan. In 2008, RQ was officially given the scientific name *Chenopodium formosanum* Koidz. (*Crook, 2021*). Characteristically, RQ exhibits a high protein (14%) and dietary fiber (14%) content as well as contains various vitamins, minerals, essential amino acids, betalains, flavonoids (such as rutin and vitamin P) (*Ando et al., 2002*; *Lin et al., 2019*), saponins, and phenolic acids (*Gómez-Caravaca et al., 2011*).

Rutin exhibits anti-inflammatory (*Woldemichael & Wink, 2001*), antioxidative (*Yao, Shi & Ren, 2014*), antitumor, anti–allergic rhinitis (*Estrada, Li & Laarveld, 1998*), and hepatoprotective (*Gómez-Caravaca et al., 2011*) bioactivities. Moreover, RQ has been reported to have strong antioxidative (*Wang, Mao & Wei, 2012*; *Lin et al., 2019*), anti-inflammatory, antidiabetes (*Li et al., 2021*), skin-protective (*Delatorre-Herrera, 2003*), and hepatoprotective (*Delatorre-Herrera, 2003*; *Lin et al., 2019*) activities. In particular, the ethanol extract of RQ bran has been noted to exhibit a strong hepatoprotective and antifibrosis effects (*Lin et al., 2019*). RQ also has been noted to have a chemopreventive effect against carcinogen-induced colon carcinogenesis through antioxidative and apoptotic pathway regulation in rats (*Lee et al., 2019*). Recently, polysaccharide bioactivities—including free radical scavenging, lipid oxidation inhibition, natural killer (NK) cell cytotoxicity promotion, macrophage activation, interleukins activity stimulation (*Song et al., 2010*; *Wang et al., 2004*; *Yang, Zhao & Lv, 2008*; *Zhu & Qian, 2006*), and antioxidative bioactivity (*Hu et al., 2017*)—have gained considerable attention. The antioxidant and immunoregulatory activities of the polysaccharides from *Chenopodium quinoa* Willd. (CQ) were evaluated by *Yao, Shi & Ren (2014)*, and the polysaccharides from CQ seeds have been reported to have antitumor activity (*Hu et al., 2017*).

Many methodologies for polysaccharide extraction from CQ and other *Chenopodium* sp. grains have been reported (*Liu et al., 2018*; *Shi, 2016*); they include water extraction, alkaline extraction (*Chen et al., 2014*), and ethanol precipitation. Of these, alkaline extraction may destroy the 3D structure of polysaccharides, leading to loss of bioactivity, and water extraction and ethanolic precipitation, although cost-effective, are time-consuming. Fermentative liquid extraction has advantages including simple operation, short production cycle, high yield, and low energy consumption; hence, it is widely considered to be feasible for industrial mass production (*Liu et al., 2018*). However, this method may cause some nutritional loss and destruction of polysaccharide structure. In addition, some adjuvant technologies such as ultrasonic and microwave-assisted extraction have been frequently applied, yet up to present to increase yield; nevertheless, water extraction and ethanolic precipitation remain the most commonly used methods (*Liu et al., 2018*; *Shi, 2016*; *Hu et al., 2017*).
Notably, Taiwanese aboriginal peoples have been consuming RQ through food and drink for a long time; however, the nutritional value of RQ grown in Taiwan remains unclear. Moreover, the characteristics of RQ polysaccharides have seldom been reported, specifically those of bioactive polysaccharides in RQ bran. In this study, we delineated some crucial nutrients occurring in RQ bran. To our knowledge, this is the first study to investigate the potent antioxidant soluble polysaccharide fraction in the RQ bran.

## MATERIALS & METHODS

RQ grains harvested in 2018 were gifted to us by the Taitung Farmer's Association in Taiwan.

### Chemicals

Hydrogen peroxide ($H_2O_2$), sodium dihydrogen phosphate ($NaH_2PO_4$), and disodium hydrogen phosphate were obtained from Taiwan Green Version Technology. Sodium hydroxide, potassium hydroxide, hydrochloric acid, sodium chloride, cupric sulfate, potassium sulfate sulfuric acid, and acetic acid EP were procured from Shimakyu (Osaka, Japan). Dialysis tubing was purchased from Fujifilm Wako Pure Chemicals (Osaka, Japan), and a Bradford Protein assay kit from Bio-Rad (Taiwan, Taipei). All other chemicals, unless otherwise indicated, were from Sigma-Aldrich (St. Louis, MO, USA).

### Proximate composition analysis

Ascorbic acid content was analyzed according to AOAC 967.21 (1968), whereas moisture, ash, fat, and protein contents were determined using the ICC standard procedure (*ICC, 1995*). In brief, moisture content was determined in an oven at 130 °C, and ash was quantified using a muffle furnace at 525 °C. Protein content was determined using the Kjeldahl method (N × 6.25), and lipid content was determined using the Soxhlet method. Crude fiber content was determined according to AOAC Official Method 962.09, and that of ascorbic acid was performed according to AOAC 967.21 (1968). The carbohydrate content was calculated as follows:

$$\%\text{Carbohydrate} = 100 - (\%\text{moisture} + \%\text{lipids} + \%\text{proteins} + \%\text{crude fiber} + \%\text{ash}). \quad (1)$$

### Total phenolic content determination

We used the method of *Taga & Miller (1984)* for determining the total phenolic contents. In brief, desiccated RQ grains (0.5 g) were transferred into a 250-mL glass flask, to which 30 mL of 0.12% HCl–methanol solution (60:40) was added, followed by ultrasonication for 30 min, after which the solution was left at ambient temperature for 2 h with constant stirring. The mixture was centrifuged at 3,000× g for 30 min, and the supernatant was collected. The residue was repeatedly extracted two more times. The three supernatants were combined and made up to a total volume of 100 mL by using the HCl–methanol solution. Three separate 0.1-mL aliquots of this mixture were then mixed with two mL of 2% $Na_2CO_3$ with vigorous agitation for 5 min. The solution was then allowed to stand at

ambient temperature for 2 min. Next, 0.1 mL of 50% Folin–Ciocalteu reagent was added, and the mixture was agitated vigorously and allowed to stand at ambient temperature for 30 min. The optical density (OD) was finally measured at 750 nm.

## Betalain content determination

As indicated by *Khatabi et al. (2011)*, 25 mg of desiccated RQ grain powder was measured and transferred into a 250-mL glass flask; to this, 30 mL of methanol was added, and the mixture was ultrasonicated for 30 min. Extraction was performed at ambient temperature for 2 h in dark. The mixture was centrifuged at $3,000\times$ g for 30 min, and the supernatant was aspirated. The residue was repeatedly extracted twice. All the three supernatants were combined and made up to total volume of 100 mL with methanol. The extract was agitated to a homogeneous state and filtered through a syringe disc filter; the OD of the filtrate was measured at respective maximum absorption wavelengths at 482 nm(for betaxanthin, solution A) and 532 nm (for indicaxanthin, solution B). The molar extinction coefficients for betaxanthin and indicaxanthin are 62,000 and 48,000 L mol$^{-1}$ cm$^{-1}$, respectively (*Girod & Zryd, 1991*). The results are expressed as milligrams of pigment per 100 g of grains.

## Free fatty acid content determination

The method by *Folch, Lees & Stanley (1957)* was followed with slight modification for determining the free fatty content in the RQ grains. In brief, 3 g of desiccated RQ grain powder was added to a 250-mL flask; to this, 50 mL of chloroform–methanol (2:1) solvent was added, which was followed by ultrasonication for 30 min. The mixture was left to stand at ambient temperature for 1 h with constant stirring and then filtered with suction; the filtrate was then transferred into a separatory funnel. Next, five mL of 0.1 N NaCl was added, followed by vigorous agitation. The stopper was released, and the mixture was left to stand for separation. Then, 10 mL of methanol was added to facilitate phase separation. The lower layer was separated and made up to a total volume of 60 mL with additional mixed chloroform:methanol (2:1); this mixture is termed the crude fat extract (CFE) hereafter. A 0.5-mL aliquot of CFE was transferred into a 2-mL reactor and dried through nitrogen blowing. To the desiccated residue, 0.1 mL of n-hexane, 0.1 mL of internal standard (*i.e.,* C15 standard acid), and 0.4 mL of BF3–methanolic solution was added; this mixture was heated at 100 °C for 30 min to facilitate the derivation reaction. While heating, samples were collected every 10 min and agitated to facilitate a homogeneous reaction. After cooling down the mixture, we added 0.1 mL of deionized water and one mL of n-hexane, agitated it for 3 min, and left it to stand to facilitate phase separation. The upper layer was collected, and a large excess of anhydrous sodium sulfate was added to it for complete dehydration. The final dehydrated oily sample was subjected to GC analysis with GC-2010AFAPC, 230V (Shimadzu, Japan). A CP-Sil 88 capillary column ($\ell \times$ i.d. $= 100$ m $\times$ 0.25 mm, inner membrane thickness $= 0.2$ m) was used. The operational temperatures were set as follows: 250 °C at the injection port and 300 °C at the detector, with the temperature elevation rate set at 3 °C/min starting from 170 °C to 200 °C within 50 min. The mobile phase was nitrogen gas controlled at a flow rate of 0.75 mL/min. The amount of sample for injection was 1 L.

## Extraction of soluble polysaccharide fractions from RQ bran

### Hot water extraction

We transferred 100 g of desiccated RQ grain hulls into a 3-L flask and then added 2 L of distilled water; this was followed by ultrasonication for 30 min and heating at 90 °C for 2 h. The mixture was then left to cool at 4 °C overnight and centrifuged at 10,000× g for 30 min. This extraction was repeated two times, and the obtained supernatants were combined to obtain approximately 6 L of hot-water extract (S1). S1 was lyophilized to obtain the first polysaccharide fraction CF-1, which was stored at 4 °C until further use (Fig. 1). The remaining residue (R1) was further treated with 2% NaOH solution.

### 2% NaOH extraction

R1 was transferred into a 3-L flask, mixed with 2 L of 2% NaOH, ultrasonicated for 30 min, heated at 80 °C for 2 h, left to cool at 4 °C overnight, and finally, subjected to centrifugation at 10,000× g for 30 min. This extraction was repeated two times. Both the supernatants were combined to obtain S2 (approximately 6 L of 2% NaOH extract; Fig. 1). The remaining residue (R2) was extracted further with 10% KOH.

### 10% KOH extraction

R2 was transferred into a 3-L flask, mixed with 2 L of 10% KOH, ultrasonicated for 30 min, heated at 80 °C for 2 h, cooled at 4 °C overnight, and finally, subjected to centrifugation at 10,000× g for 30 min. This extraction was repeated two times. Both the supernatants was to obtain S3 (approximately 6 L of 10% KOH extract; Fig. 1) were combined for use.

### Isoelectric point precipitation

The extract S2 (6 L of 2% NaOH extract) was adjusted with concentrated $H_2SO_4$ to an isoelectric pH of 4.0, stored at 4 °C overnight to facilitate precipitation, and centrifuged at 10,000× g for 30 min to obtain the residue R4 and the supernatant S4. R4 was lyophilized to obtain the polysaccharide fraction CF-2 (Fig. 1). S3 (6 L of 10% KOH extract) was similarly treated with isoelectric precipitation to obtain supernatant S5; the residue R5 was discarded (Fig. 1).

### Ethanolic precipitation

S4 and S5 were separately redissolved in 500 mL of distilled water and reprecipitated by adding 3 to 5-fold (v/v) 95% ethanol and leaving it to stand for 24 h in 4 °C. The mixture was centrifuged at 10,000× g for 30 min, and the supernatant was discarded. The precipitates were dialyzed for 48 h and then lyophilized to obtain CF-3 and CF-4 from S4 and S5, respectively (Fig. 1).

## Gel permeation chromatography

By using the method of *Wang, Mao & Wei (2012)* with slight modification, we performed gel permeation chromatography on the soluble polysaccharide fractions. In brief, we added 10 mg of each fraction to separate 20-mL sampling vials. Then, one mL of 1 N NaOH solution was added, and the mixtures were ultrasonicated for 30 min to facilitate dissolution. Next, we added two mL of distilled water and then vigorously agitated the vials. A 0.5-mL aliquot was injected into a Superdex G-200 gel column

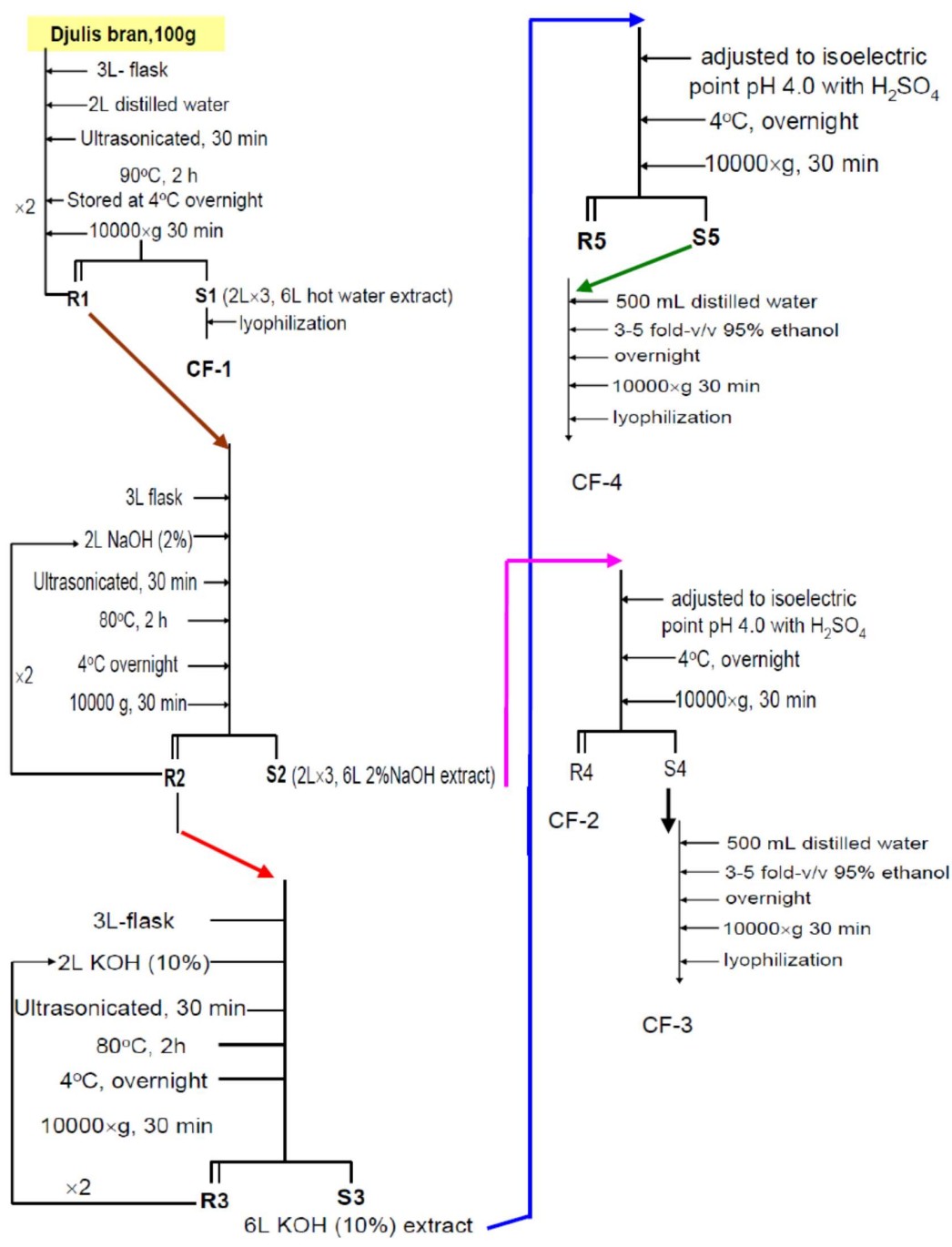

**Figure 1** The laboratory systemic protocol for isolation and purification of polysaccharides from the brans of *C. formosanum* Koidz.

($\ell \times d = 76$ cm $\times$ 1.6 cm), with mobile phase flow rate = 0.3 mL/min. The eluent was collected using a fraction collector in 50 tubes (two mL per tube). The OD of the main portion of the eluents in the tubes with odd number was measured at 280 nm. Next, a part

of the eluent (~0.5 mL of eluent from each tube) was subjected to the phenol–sulfuric acid color reaction; the resulting OD was measured at 490 nm.

## Determination of physicochemical parameters in polysaccharide fractions

### Carbohydrate content

As reported by *Dubois et al. (1956)*, 5 mg of each polysaccharide fraction was transferred to a 20-mL sample vial, mixed with four mL of 1 N NaOH through ultrasonication for 30 min, and heated at 50 °C for 2 h with constant stirring. The solution was made up to a total volume of five mL with 1N NaOH and mixed well with agitation. To a 0.5-mL aliquot of this mixture, we added 0.5 mL of phenol with agitation and mixed the solution well; this was followed by the addition of 2.5 mL of concentrated $H_2SO_4$ and thorough mixing. The solution was then left to stand until a consistent orange yellow coloration appeared. Its OD was measured at 490 nm, next. The experiment was performed in triplicate. Moreover, standard glucose was used to establish a calibration curve, which was used to calculate the amount of carbohydrate.

### Peptide content

As per the method described by *Bradford, (1976)*, 5 mg of each polysaccharide fraction was transferred to a 20-mL sample vial, mixed with four mL of 1N NaOH through ultrasonication for 30 min, and heated at 50 °C for 2 h with constant stirring. The solution was made up to a total volume of 5 mL with 1N NaOH and mixed well with agitation. To a 0.1-mL aliquot of this mixture, we added two mL of the Dye Reagent Concentrate and mixed well; the solution was left to stand until a stable bluish brown color appeared. Its OD was measured at 595 nm. This experiment was also performed in triplicate. Standard bovine serum albumin (BSA) was used to establish a calibration curve, which was used for calculating the amount of protein.

### Monosaccharide composition

Here, we followed the method of *Blakeney et al. (1983)* with a slight modification. In brief, 3 mg of each polysaccharide fraction was transferred into a 3-mL sample vial. To this, we added three mL of 2 M trifluoroacetic acid (TFA) and followed it by ultrasonication for 30 min. The solution was then transferred into a derivatizer and heated at 120 °C for 24 h (with being removed and the solution being agitated every hour) until complete hydrolyzation was achieved. The reaction solution was then mixed with three mL of deionized water (d.w.) in a 20-mL sample vial, followed by agitation and lyophilization. To the lyophilized residue, we added 0.5 mL of d.w. and then agitated the mixture until the residue dissolved completely (denoted as the completely acid hydrolyzed solution from the polysaccharides, AHP). A 100-L aliquot of AHP was transferred to a 3-mL reaction vial and mixed with 100 μL of 0.6 M NaOH with agitation; next, 200 μL of 0.5 M PMP methanolic solution was added to this. After thorough agitation, the solution was transferred to the derivatizer heated at 70 °C for 2 h to facilitate the derivation reaction with agitation every 30 min. After cooling the mixture at ambient temperature, it was mixed with 200 μL of 0.3 M HCl was for neutralization. The cap of the derivatizer was removed and placed onto a derivatizer heater

and then heated at 100 °C for evaporation until completely dry. Then, one mL of d.w. was added and mixed well through agitation. Next, one mL of chloroform was added, followed by agitation for 3 min, and the mixture was allowed to stand at ambient temperature to facilitate phase separation. The upper layer was then separated. This extraction was repeated two more times. The three upper layers were combined and subjected to high-performance liquid chromatography (HPLC). The HPLC system (Shimadzu, Japan) used was equipped with a C18ACE reverse phase column ($\ell$ × i.d. = 250 mm × 4.60 mm, thickness 5 μm) maintained at 25 °C, an LC-20AT liquid chromatography pump (100–120 V, 50–60 Hz, 150 VA), and an SPD-20A UV–visible detector ($\sim$120 V, 50–60 Hz, 160 VA), and it was operated with 0.1 M $KH_2PO_4$ buffer (pH 6.7): acetonitrile (83:17 v/v) as the mobile phase at a flow rate of one mL/min as well as a detection wavelength of 245 nm and sample injection volume of 20 L.

### Amino acid composition of peptide moiety

Here, we use the method of *Namera et al. (2002)* with slight modifications. In brief, 5 mg of each polysaccharide fraction was transferred to a 3-mL reaction vessel and mixed with three mL of 6 M HCl; this vessel was placed onto a derivatizer heated at 110 °C for 24 h with agitation every hour. Next, the mixture was cooled and lyophilized to obtain a powder. This powder was mixed with 0.3 mL of 0.01 M HCl solution until dissolution. The mixture was then transferred into a 3-mL derivatizer vial and mixed with 0.01 mL of 10 mg/mL norleucine (the internal standard), followed by agitation, addition of 0.05 mL of ethyl chloroformate, and agitation again. We then added 0.5 mL of alcohol–pyridine and one mL of chloroform and agitated the solution for 1 min to facilitate the derivation and the next extraction. The mixture was left to stand at ambient temperature for 5 min. Next, we added 0.7 mL of distilled water to the solution, agitated it for 30 s, and left it to stand for another 5 min to facilitate phase separation. The upper layer was discarded, whereas the lower layer (*i.e.,* the chloroform layer) was collected and dehydrated with anhydrous sodium sulfate. Next, only the upper clear liquid was collected and transferred into a sample vial, concentrated to only a minute amount with the aid of nitrogen blowing. The concentrated sample was subjected to gas chromatography (GC) on a GC-2010AFAPC, 230V (Shimadzu, Japan) equipped with a HP-5MS column ($\ell$ × i.d. = 30 m × 0.25 mm; film thickness = 0.25 μm) and a flame ionization detector (FID). The column temperature was programmed as follows: hold at 50 °C for 1 min, followed by temperature elevation at 10 °C/min until 300 °C, and then, hold for 6.5 min. The injection port temperature was set at 300 °C, and the sample size at 1 L. The sensor temperature was set at 305 °C. $N_2$ gas was the mobile phase, controlled at a flow rate of one mL/min.

## Determination of antioxidative capability of polysaccharide fractions
### 1,1-Diphenyl-2-picrylhydrazyl free radical–scavenging capability

We followed the method of *Thaipong et al. (2006)* to assay for 1,1-diphenyl-2-picrylhydrazyl (DPPH) free radical–scavenging capability (FRSC) of our fractions. In brief, 100 mg of each polysaccharide fraction was transferred to a sample vial. Next, eight mL of 0.01 M phosphate buffer (pH 7.4) was added, followed by ultrasonication for 30 min and heating at 50 °C for 24 h with constant stirring until complete dissolution. The solution was

centrifuged at 3,000× g for 30 min. The supernatant was separated and made up to a total volume of 10 mL with 0.01 M phosphate buffer (pH 7.4; denoted as the stock solution). The stock solution was diluted with 0.01 M phosphate buffer (pH 7.4) to prepare sample solutions of different concentrations: 10, 5, 1, 0.5, 0.1, 0.05, and 0.01 mg/mL. Every 0.8-mL aliquot of each sample was mixed well with 0.2 mL of 0.5 mM DPPH methanolic reagent, mixed well, and then left to stand in dark for 30 min. The OD was measured at 517 nm ($A_s$) against a blank (0.01 M phosphate buffer, pH 7.4; $A_b$). The experiment was repeated three times. The DPPH FRSC was calculated using the following equation:

$$\%_{DPPH\ free\ radical\ scavenging} = [1 - (A_s/A_b)] \times 100. \tag{2}$$

### Hydroxyl free radical inhibitory activity

As indicated by *Ghiselli, (1998)*, 100 mg of each polysaccharide fraction was transferred to a sample vial and mixed with eight mL of 0.01 M phosphate buffer (pH 7.4) through ultrasonication for 30 min and heating at 50 °C for 24 h with constant stirring until complete dissolution. The solution was then centrifuged at 3,000× g for 30 min. The supernatant was separated and made up to a total volume of 10 mL with 0.01 M phosphate buffer (pH 7.4; denoted as the stock solution). The stock solution was then diluted with 0.01 M phosphate buffer (pH 7.4) to prepare sample solutions of different concentrations: 10, 5, 1, 0.5, 0.1, 0.05, and 0.01 mg/mL. Next, a 0.1-mL aliquot of each of these solutions was mixed with 0.6 mL of 0.0025 M deoxyribose phosphate (in 0.01 M phosphate buffer, pH 7.4) and agitated for uniform mixing. To the solution, 0.2 mL of d.w. containing 1.04 mM EDTA and 0.1 mM $Fe_2(NH_4)_2(SO_4)_3$ was added. For each sample, a 0.1-mL aliquot was mixed with 50 μL of 2 mM vitamin C through agitation; next, 50 μL of 0.1 M $H_2O_2$ was added, and the solution was mixed well, capped, and left to react at ambient temperature for 30 min. We then added one mL of 0.02 M TBA and one mL of 2% TCA and followed it by agitation, capping, and heating in boiling water for 15 min until the color turned pink. The solution was left to stand at ambient temperature for 10 min. The OD was measured at 532 nm ($A_{sample}$) against a blank 0.01 M phosphate buffer (pH 7.4; $A_{control}$). This experiment was performed in triplicate. The % hydroxyl free radical inhibitory capability was calculated as follows:

$$\begin{aligned}\%\bullet OH_{inhibitory\ capability} \\ = [1 - (A_{sample,532nm}/A_{control,532nm})] \times 100.\end{aligned} \tag{3}$$

### Superoxide FRSC

Here, we used *Nishikimi, Rao & Yagi (1972)* method with a slight modification. In brief, 100 mg of each polysaccharide fraction was mixed with eight mL of 0.01 M phosphate buffer (pH 7.4) in a 10-mL sample vial through ultrasonication for 30 min and heating at 50 °C for 24 h with constant stirring until complete dissolution—followed by centrifugation at 3,000× g for 30 min. The supernatant was collected and made up to a total volume of 10 mL with 0.01 M phosphate buffer (pH 7.4; denoted as the stock solution). The stock

solution was diluted with 0.01 M phosphate buffer (pH 7.4) to prepare sample solutions of different concentrations: 10, 5, 1, 0.5, 0.1, 0.05, and 0.01 mg/mL. Next, a 0.25-mL aliquot of each of these solutions was mixed with two mL of 0.01 M phosphate buffer (pH 7.4; containing 0.1 mM EDTA, 0.062 mM nitro blue tetrazolium, and 0.098 mM NADH) with agitation. Next, 0.25 mL of phosphate buffer (pH 7.4; containing 0.045 mM phenazine methosulfate) was mixed in well, and the mixture was allowed to stand at ambient temperature for 5 min until a bluish purple color formed. The OD was read at 560 nm against a blank (0.01M phosphate buffer, pH 7.4) solution. This experiment was performed in triplicate. The % superoxide FRSC was calculated as follows:

$$\% \bullet O_2^- \ scavenging \ capability = [1 - (A_{sample,560nm}/A_{control,560nm})] \times 100. \tag{4}$$

### Ferrous ion–chelating capability

The method of *Saito & Ishihara (1997)* was followed with slight modification for this assay. Briefly, 100 mg of each polysaccharide fraction was mixed with eight mL of 0.01 M phosphate buffer (pH 7.4) in a 10-mL sample vial through ultrasonication for 30 min and heating at 50 °C for 24 h with constant stirring to facilitate complete dissolution and then centrifugation at $3,000 \times$ g for 30 min. The supernatant was separated and made up to a total volume of 10 mL with 0.01 M phosphate buffer (pH 7.4; denoted as the stock solution). The stock solution was diluted with 0.01 M phosphate buffer (pH 7.4) to prepare sample solutions of different concentrations: 10, 5, 1, 0.5, 0.1, 0.05, and 0.01 mg/mL. Then, a 0.1-mL aliquot of each of these sample solutions was mixed well with 1.85 mL of d.w. and 0.25 mL of 2 mM $FeCl_2 4H_2O$ and then left to stand for 30 s at ambient temperature; next, 0.25 mL of 5 mM ferrozine was added and agitated for uniform mixing. The solution was kept in the dark for 10 min, and then its OD was measured at 562 nm against the blank (0.01 M phosphate buffer, pH 7.4). This experiment was performed in triplicate. The % ferrous ion chelating capability was calculated as follows:

$$\% Fe^{2+}_{chelating \ capability} = [1 - (A_{sample,562nm}/A_{control,562nm})] \times 10. \tag{5}$$

## Fourier transform infrared spectrometry analysis

The polysaccharide fractions that showed the highest antioxidative capability were selected and subjected to Fourier transform infrared spectrometry (FTIR) analysis, performed according to the manufacturer's instruction; the obtained spectra were scanned using Jusco FT/IR-460 within 4,000–7,000 $cm^{-1}$.

## Statistical analysis

All experiments were conducted in triplicate, and all statistical analysis were performed in SPSS by using analysis of variance (ANOVA)within the same group. Moreover, the Duncan multiple-range test was used to compare means among groups. Significant difference was indicated by a $p$ of $<0.05$.

**Table 1** The fatty acid composition of crude fat isolated from the bran of *C. formosanum* Koidz.

| Fatty acid Saturated fatty acids | % (w/w) | Fatty acid Unsaturated fatty acids | % (w/w) |
|---|---|---|---|
| C8:0 | 0.80 | C16:1;9 | 0.39 |
| C10:0 | 12.05 | C18:1;9 | 4.74 |
| C12:0 | 8.82 | C18:2;9,12 | 22.72 |
| C13:0 | 0.52 | C18:3;9,12,15 | 13.52 |
| C14:0 | 1.09 | C20:3;8,11,14 | 3.00 |
| C16:0 | 21.21 | C22:1;13 | 2.82 |
| C17:0 | 1.31 | Total | 47.19 |
| C18:0 | 3.63 | | |
| C20:0 | 0.67 | | |
| C22:0 | 1.05 | Monounsaturated fatty acids | 7.95 |
| C24:0 | 1.66 | Polyunsaturated fatty acids | 39.24 |
| Total | 52.81 | | |

# RESULTS

## Proximate composition

RQ was found characteristically to contain an abundant amount of crude protein (16.56% ± 0.43%), crude fiber (6.14% ± 0.04%), and ash (7.10% ± 0.06%) and a moderate level of crude fat (4.06% ± 0.06%) and carbohydrate (60.45% ± 0.76%). It also contained high total phenolic content (1.95% ± 0.08%) and betalains (*i.e.,* 9.91% ± 0.22% betaxanthin and 7.27% ± 0.32% indicaxanthin). However, the RQ has been reported to contain 2.36–4.27 mg/g betacyanins and 1.84–3.32 mg/g betanins+isobetanins (*Tsai et al., 2011*).

## Fatty acid content and profile

Crude fat from RQ bran contained 52.81% (w/w) total saturated fatty acids (SFAs) and 47.19% (w/w) unsaturated fatty acids. Moreover, it contained 7.95% (w/w) monosaturated fatty acids and 39.24% (w/w) polyunsaturated fatty acids (PUFAs; Table 1). Palmitic acid (C16) content was the highest (21.21% w/w), followed by that of lauric acid (C12) (8.82% w/w). By contrast, stearic acid (C18) content was the lowest at only 3.63% (w/w; Table 1). Notably, the crude fat also contained odd C-numbered (C13 and C17) SFAs: 0.52% (w/w) C13 tridecylic acid (or tridecanoic acid) and 1.31% (w/w) C17 margaric acid (or heptadecanoic acid; Table 1).

In addition, the crude fat contained unique SFAs such as C22-behenic acid (docosanoic acid; 1.05%) and C24-lignoceric acid (tetracosanoic acid; 1.66% w/w; Table 2). Of the unsaturated fatty acids, linoleic acid (C18:2;9,12) and linolenic acid (C18:3;9,12,15) contents were the highest, reaching 22.72% and 13.52%, respectively (Table 1). C20-eicosatrienoic acid (C20:3;8,11,14), another unique fatty acid, was also detected; its structure was completely different from that of the common omega-3 C20-eicosatrienoic acid (C20:3;11,14,17), which has been reported frequently. The crude fat however appeared to lack arachidonic acid (eicosatetraenoic acid, C20:4;5,8,11,14; Table 1).

**Table 2** Yields, mean molecular weight, carbohydrate moiety, and peptide moiety of polysaccharide fractions isolated from the bran of *C. formosanum* Koidz[*].

| Polysaccharide fraction | Weight (g) | Yields (%) | Carbohydrate moiety (%) | Peptido moiety (%) | MW (kDa) |
|---|---|---|---|---|---|
| CF-1 | 3.9065 | 3.90 | $38.92 \pm 0.38^c$ | $15.67 \pm 0.48^b$ | 32.54 |
| CF-2 | 6.7469 | 6.74 | $50.70 \pm 0.59^b$ | $42.41 \pm 0.93^a$ | 24.93 |
| CF-3 | 22.2933 | 22.28 | $93.76 \pm 0.72^a$ | $5.44 \pm 0.20^c$ | 72.39 |
| CF-4 | 0.0621 | 0.06 | $19.80 \pm 0.94^d$ | $14.52 \pm 0.33^b$ | 55.45 |

Notes.
*CF-1: the three-fold ethanol precipitate from hot water extracts. CF-2: the isoelectric precipitate from the 2%-NaOH extracts. CF-3: the three-fold ethanol precipitate from the 2%-NaOH extracts post isoelectric precipitation. CF-4: the three-fold ethanol precipitate from the 10%-KOH extracts post isoelectric precipitation. MW: mean MW from the gel filtration chromatography.*Values are expressed as mean $\pm$SD from triplicate determinations ($n = 3$). Means within the same column with different superscripts in lower case are significantly different ($p < 0.05$).

## Physicochemical properties of isolated polysaccharides

In CF-1, CF-2, CF-3, and CF-4, the polysaccharide content was 3.9%, 6.74%, 22.28%, and 0.06%, respectively; the carbohydrate content was 38.92%, 50.70%, 93.76%, and 19.80%; and the peptide moiety content was 15.67%, 42.41%, 5.44%, and 14.52%, respectively (Table 2, Figs. S1A–S1D). CF-3 had the largest molecular weight (MW; 72.39 kDa), followed by CF-4 (55.45 kDa), CF-1 (32.54 kDa), and finally, CF-2 (24.93 kDa; Table 2, Figs. S1A–S1D).

## Monosaccharide types and contents in each isolated polysaccharide

Different polysaccharide fractions had different monosaccharides with varying contents. CF-2 and CF-4 contained mannose, glucuronic acid, ribose, rhamnose, galacturonic acid, glucose, galactose, and xylose, whereas CF-1 and CF-3 did not contain any ribose.

CF-1 contained glucose (60.91 mol %; peak 4), galactose (10.97 mol %; peak 5), xylose (10.08 mol %; peak 6), mannose (6.69 mol %; peak 1), glucuronic acid (4.66 mol %), rhamnose (4.45 mol %; peak 3), and galacturonic acid (2.24 mol %; Table 3, Fig. S2A).

CF-2 contained glucose (95.86 mol %; peak 2), galactose (0.96 mol %; peak 3), rhamnose (0.96 mol %; peak 1), mannose (0.64 mol %), galacturonic acid (0.51 mol %), ribose (0.19 mol %), and glucuronic acid (0.11 mol %; Table 3, Fig. S2B).

CF-3 contained glucose (87.24 mol %; peak 2), xylose (5.07 mol %; peak 4), rhamnose (3.21 mol %; peak 1), galactose (3.11 mol %; peak 3), and other minor monosaccharides (<1.0 mol %; Table 3, Fig. S2C).

CF-4 contained glucose (85.37 mol %; peak 2), galacturonic acid (5.73 mol %; peak 1), galactose (4.96 mol %; peak 3), xylose (2.11 mol %; peak4), and other minor monosaccharides (<1.0 mol %; Table 3, Fig. S2D).

## Amino acid types and contents in each isolated polysaccharide

In total, 15 major amino acid types were detected in the four polysaccharide fractions. CF-1 contained alanine (18.13%), leucine (16.90%; peak 3), valine (13.85%; peak 2), phenylalanine (12.44%; peak 5), cysteine (10.91%; peak 6), isoleucine (9.26%; peak 4), and glycine (0.37%; peak 1; Table 4, Fig. S3A); CF-2 contained leucine (16.23%; peak 3), phenylalanine (15.75%; peak 5), isoleucine (12.55%; peak 4), valine (10.79%; peak 2),

**Table 3** The mole percentage of monosaccharides in different polysaccharide fractions obtained isolated from the bran of *C. formosanum* Koidz[*].

| Monosaccharide (mole%) | Polysaccharide fraction | | | |
|---|---|---|---|---|
| | CF-1 | CF-2 | CF-3 | CF-4 |
| Mannose | 6.69 | 0.64 | 0.31 | 0.97 |
| Glucuronic acid | 4.66 | 0.11 | 0.27 | 0.29 |
| Ribose | n.d.[#] | 0.19 | n.d | 0.25 |
| Rhamnose | 4.45 | 0.96 | 3.21 | 0.46 |
| Galacturonic acid | 2.24 | 0.51 | 0.80 | 5.73 |
| Glucose | 60.91 | 95.86 | 87.24 | 85.37 |
| Galactose | 10.97 | 0.96 | 3.11 | 4.96 |
| Xylose | 10.08 | 0.77 | 5.07 | 2.11 |

**Notes.**
[#]n.d.: not detected.
[*]CF-1: the three-fold ethanol precipitate from the hot water extracts. CF-2: the isoelectric precipitate from the 2%-NaOH extracts. CF-3: the three-fold ethanol precipitate from the 2%-NaOH extracts post isoelectric precipitation. CF-4: the three-fold ethanol precipitate from the 10%-KOH extracts post isoelectric precipitation.

glycine (9.06%; peak 1), cysteine (6.21%), alanine (5.69%), aspartic acid (5.51%), histidine (5.53%), and tyrosine (3.02%; Table 4, Fig. S3B); and CF-3 contained isoleucine (23.22%; peak 4), leucine (11.94%; peak 3), valine (11.15%; peak 2), aspartic acid (11.00%), phenylalanine (10.98%; peak 5), glycine (10.95%; peak 1), histidine (5.49%), alanine (4.46%), tyrosine (2.86%), and lysine (2.48%; Table 4, Fig. S3C). Notably, CF-4 contained the largest tyrosine content (15.61%; peak 5) among all the fractions, followed by leucine (14.08%; peak 3), alanine (14.03%), isoleucine (13.02%; peak 4), valine (9.19%; peak 2), aspartic acid (9.16%), histidine (5.44%), glycine (5.20%; peak 1), lysine (4.02%), proline (3.25%), cysteine (2.53%), and phenylalanine (2.88%; Table 4, Fig. S3D). Consequently, we considered CF-1, CF-2, CF-3, and CF-4 to be an alalinoglucan, leucinoglucan, isoleucinoglucan, and tyrosinoglucan, respectively (Table 3 and Table 4).

## DPPH FRSC
Because of depletion, CF-4 was not included in the subsequent experiments. CF-1, CF-2, and CF-3 demonstrated moderate but dose-dependent DPPH FRSC (Fig. 2A). At $\leq 5$ mg/mL, CF-2 has the highest capability (62.98% ± 0.74%; Fig. 2A). Even at 10 mg/mL, CF-2 had a much higher capability (87.85 ± 0.76) than did CF-1 (40.69 ± 0.41) and CF-3 (45.32 ± 0.59)—all compared with 1 mg/mL Trolox (Fig. 2A). According to *Tsai et al. (2011)*, the DPPH FRSC of RQ can be 46.42% to 65.68% depending on the finished particle sizes.

## Hydroxyl free radical inhibitory activity
CF-1, CF-2, and CF-3 demonstrated only moderate hydroxyl free radical inhibitory activity (Fig. 2B), with CF-2 demonstrating the highest activity. At 5 mg/mL, CF-2 performed 55.76% ± 0.52% inhibition, whereas at 10 mg/mL, it showed more inhibition (69.89% ± 1.17) than CF-1 (47.53% ± 0.06%) and CF-3 (42.63% ± 1.06%; Fig. 2B). At 10 mg/mL, CF-1's activity (66.95% ± 0.41%) was comparable to that of 1 mg/mL Trolox (Fig. 2B).

**Table 4** The amino acid composition occurring in different polysaccharide fractions isolated from the bran of *C. formosanum* Koidz[*].

| Amino Acid | Polysaccharide fraction | | | |
| --- | --- | --- | --- | --- |
| | CF-1 | CF-2 | CF-3 | CF-4 |
| | Amino acid content (%) | | | |
| Alanine | 18.13 | 5.69 | 4.46 | 14.03 |
| Glycine | 0.37 | 9.06 | 10.95 | 5.20 |
| Valine | 13.85 | 10.79 | 11.15 | 9.19 |
| Leucine | 16.90 | 16.23 | 11.94 | 14.08 |
| Isoleucine | 9.26 | 12.55 | 23.22 | 13.02 |
| Proline | 0.58 | 3.78 | 2.96 | 3.25 |
| Glutamic acid | n.d | n.d | n.d | n.d |
| Methionine | 2.47 | 1.29 | 1.34 | 1.59 |
| Aspartic acid | 8.39 | 5.51 | 11.00 | 9.16 |
| Hydroxyproline | n.d | n.d | n.d | n.d |
| Phenylalanine | 12.44 | 15.75 | 10.98 | 2.88 |
| Cysteine | 10.91 | 6.21 | 1.17 | 2.53 |
| Lysine | 4.21 | 4.59 | 2.48 | 4.02 |
| Histidine | 0.94 | 5.53 | 5.49 | 5.44 |
| Tyrosine | 1.55 | 3.02 | 2.86 | 15.61 |

**Notes.**

[*]CF-1: the three-fold ethanol precipitate from the hot water extracts. CF-2: the isoelectric precipitate from the 2%-NaOH extracts. CF-3: the three-fold ethanol precipitate from the 2%-NaOH extracts post isoelectric precipitation. CF-4: the three-fold ethanol precipitate from the 10%-KOH extracts post isoelectric precipitation.

## Superoxide FRSC

All three fractions demonstrated dose-dependent superoxide FRSC. At 0.1 mg/mL, all three fractions had higher superoxide FRSC (44.27%, 55.42%, and 46.34% for CF-1, CF-2, and CF-3, respectively) than did Trolox (even at 1 mg/mL, 38.17% $\pm$ 0.54%; Fig. 2C). At 1 mg/mL, CF-1, CF-2, and CF-3 exhibited a superoxide FRSC of 58.78%, 59.04%, and 67.22%, respectively. However, at 10 mg/mL, CF-3 has the highest FRSC (Fig. 2C). Nonetheless, fresh substrate RQ was noted to contain a large amount of antioxidant enzymes such as lysozymes, superoxide dismutase, and lipid peroxidase (*Tsai et al., 2011*).

## Ferrous ion–chelating capability

All three fractions demonstrated ferrous ion-chelating capability that was, although dose-dependent, much lower than that of EDTA (Fig. 2D). As mentioned above, the antioxidant capability of RQ depends on the particle size, and its FRAP ranges from 517.24 to 713.92 mol/L (*Tsai et al., 2011*).

## $IC_{50}$ for different antioxidant parameters

The $IC_{50}$ of each antioxidant parameter is listed in Table 5. CF-1, CF-2, and CF-3 were found to be highly effective against superoxide free radicals, with their $IC_{50}$ reaching 0.26, 0.05, and 0.19 mg/mL (Table 5). Notably, among the three fractions, CF-2 had the most effective DPPH, superoxide, and $H_2O_2$ FRSCs (Table 5).

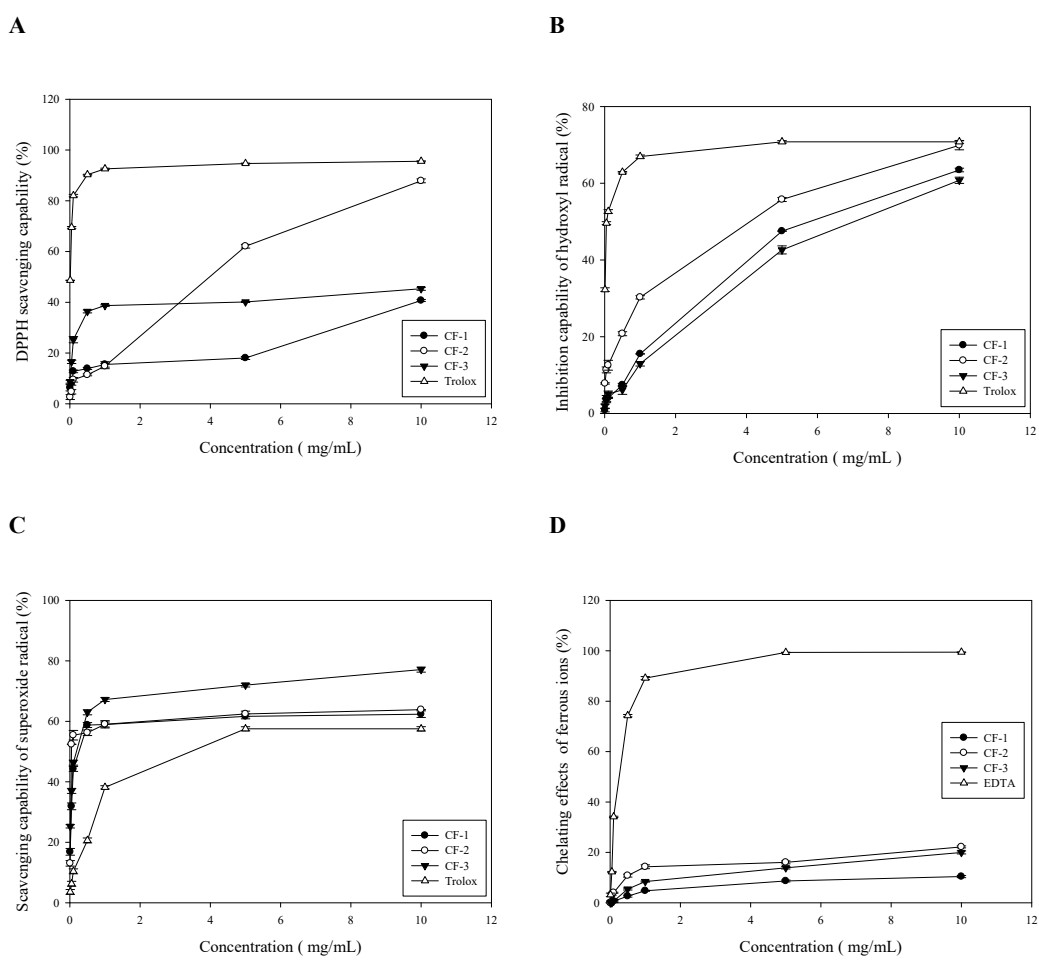

**Figure 2  The antioxidative capability of different fractions obtained from the brans of *C. formosanum* Koidz.** (A) The DPPH scavenging capability. BHA was used as a reference control. (B) The hydroxyl free radical inhibitory activity. Trolox was used as the reference control. (C) The superoxide free radicals scavenging capability. Trolox was used as the reference control. (D) The Fe2+-chelating capability. EDTA was used as the reference control. CF-1: the three-fold ethanol precipitate from the hot water extracts. CF-2: the isoelectric precipitate from the 2%-NaOH extracts. CF-3: the three-fold ethanol precipitate from the 2%-NaOH extracts post isoelectric precipitation.

## FTIR characterization of polysaccharides

The close, tight coincidence of the OD spectrum at 280 nm with that at 490 nm in Fig. 2B strongly implicated the high purity of CF-2; moreover, CF-2 demonstrated a broad spectrum of great biological properties. Thus, only CF-2 was subjected to FTIR analysis. Most polysaccharides demonstrate similar absorption bands in the region of 450–4,000 cm$^{-1}$, and these are thus called the characteristic peaks of polysaccharides (*Ren et al., 2017*; *Kolsi et al., 2017*). Here, the broad stretching peaks at 3,346–3,370 cm$^{-1}$ could be ascribed to the hydroxyl groups with stretching vibration, and the absorption bands within 2,930–2,933 cm$^{-1}$ were attributable to C–H stretching vibrations ($\nu_{\text{C-H}}$) of polysaccharides. A strong band was derived from C =O stretching vibration ($\nu_{C=O}$; caused by glucuronic

**Table 5   The estimated IC$_{50}$ values for different antioxidant capacity of each polysaccharide fraction[*].**

| Polysaccharide fraction | IC$_{50}$ values for different antioxidant parameters, mg/mL | | | |
|---|---|---|---|---|
| | DPPH scavenging | H$_2$O$_2$ scavenging | ●O$_2^-$ Scavenging | Fe$^{2+}$ chelation |
| CF-1 | 12.05 | 5.77 | 0.26 | 100.41 |
| CF-2 | 3.98 | 4.10 | 0.05 | 28.12 |
| CF-3 | 14.5 | 7.03 | 0.19 | 29.73 |

Notes.

[*]CF-1: the three-fold ethanol precipitate from the hot water extracts. CF-2: the isoelectric precipitate from the 2%-NaOH extracts. CF-3: the three-fold ethanol precipitate from the 2%-NaOH extracts post the isoelectric precipitation. For scavenging of DPPH, H$_2$O$_2$, and ●O$_2^-$, Trolox was used as the reference compound, for ferrous chelating, EDTA was used as the reference control.

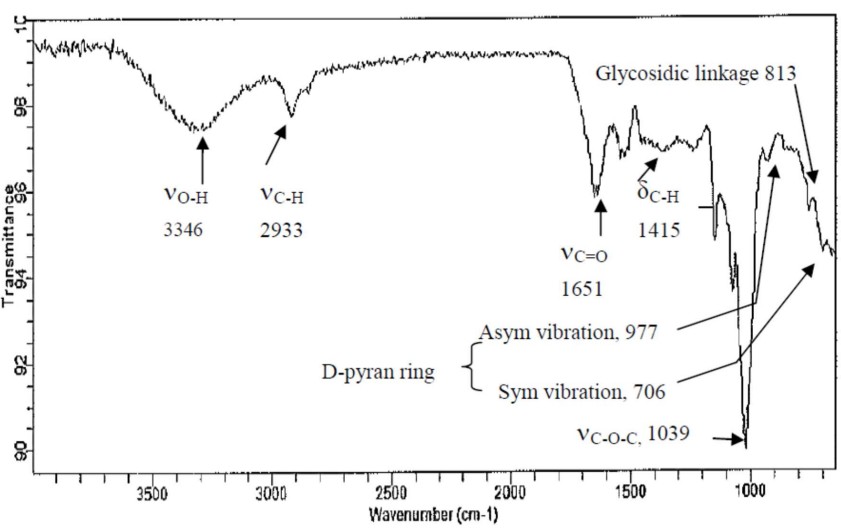

**Figure 3   FT-IR spectra of polysaccharide fractions obtained from the brans of *C. formosanum* Koidz.**

and galacturonic carboxyl groups and by bound water) at 1,651 cm$^{-1}$, and the weak band due to C–H bending vibration ($\delta_{\text{C-H}}$) was noted around 1,415 cm$^{-1}$. The 1,034–1,039 cm$^{-1}$ region demonstrated C–O–C stretching ($\nu_{\text{C-O-C}}$; the prominent band between 1,010 and 1,100 cm$^{-1}$ represented a pyran structure), suggesting the presence of pyranose sugars. Moreover, the characterization of CF-2 demonstrated the typical absorption of D-pyran ring grape asymmetric stretching vibration around 977 cm$^{-1}$ and symmetric stretching vibration around 706 cm$^{-1}$. The absorption bands within 811–813 cm$^{-1}$ were due to the a-type glycosidic linkages. Thus, CF-2 was concluded to be a pyran polysaccharide (Fig. 3).

## DISCUSSION

### High nutrient value of RQ

According to *Nascimento et al. (2014)*, 100 g of CQ contains 11.30 ± 0.05 g of moisture, 12.10 ± 0.3 g of protein, 6.31 ± 0.11 g of fat, 10.40 ± 0.60 g of fiber, 57.20 ± 0.6 g of

starch, 19.70 ± 0.5 g of amylose, and 2.01 ± 0.02 g of ash. However, *Hernández-Ledesma (2019)* presented a slightly different result: CQ is rich in in protein (16.5 g/100 g), fat (6.3 g/100 g), fiber (3.8 g/100 g), ash (3.8 g/100 g), and carbohydrates (69 g/100 g). In other literature, the nutrient composition of uncooked CQ mainly varies to a large extent in terms of proteins (16.28 g/100 g) and ash (2.74 g/100 g) (USDA, 2013 (*Wrigley et al., 2015*). By contrast, in the current study, RQ bran contained relatively high contents of crude protein (16.56% ± 0.43%), crude fiber (6.14% ± 0.04%), and ash (7.1% ± 0.06%) but moderate amounts of carbohydrate (60.45% ± 0.76%) and crude fat (4.06% ± 0.06%); it also contained abundant amounts of polyphenolics (1.95% ± 0.08%) and betalains (*i.e.,* betaxanthin, 9.91% ± 0.22 mg/100 g, and indicaxanthin, 7.27 ± 0.32 mg/100 g). Betalains have many biomedical uses because of their cardioactive, anticancer, anti-inflammatory, hepatoprotective, radioprotective, neuroprotective, diuretic, hypolipidemic, osteoarthritis pain reliever, and antidiabetic activities (*Khan, 2016*).

## Beneficial fatty acid composition of RQ

In amaranth, quinoa, and RQ, the n-6/n-3 ratios are 45.42 ± 0.82, 18.13 ± 0.05, and 18.137, respectively, and the PUFA/SFA ratios are 1.74 ± 0.02, 4.47 ± 0.05, and 0.743, respectively (*Palombini et al., 2013*; Table 1).

The *Department of Health of the United Kingdom (1994)* recommends a PUFA/SFA ratio of ≥0.1 coupled with an n-6/n-3 ratio of <4. However, in the past when the human diet was mainly composed of wild animal meat and plant matter, the ideal n-6/n-3 ratio was approximately 1 (*Cordain et al., 2005*). However, in the modern Western diet, this ratio can be unfavorably high at 10–20 (*Sanders, 2000*). By consumption more n-3 PUFAs in the daily diet, this ratio could be actively reduced, in turn improving human health greatly (*Valencak et al., 2015*).

Marine algae as well as engineered transgenic plants can be great sustainable sources of n-3 PUFAs (*Ruiz-Lopez et al., 2009*; *Martins et al., 2013*). Their incorporation of these alga-sourced diets is highly warranted in the near future.

## A typical leucinoglucan in CF-2

CF-2 was noted to contain 42.41% ± 0.93% peptide moiety (Table 4), composed of 16.23% leucine, and 50.70% ± 0.59% carbohydrate moiety (Table 2), composed of 95.86 mol % glucose (Table 3)—suggesting that CF-2 is a leucinoglucan. In addition, the tightly overlapping coincidence of the OD spectrum at 280 nm with that at 490 nm (Fig. S1B) strongly implicated the high purity of CF-2. *Rogers, Perkins & Ward (1980)* demonstrated that the main structural features of peptidoglycan are linear glycan strands crosslinked by short peptides. Our HPLC analysis revealed that in the staphylococcal glycan strands, the predominant chain length was 3–10 disaccharide units. Longer glycan strands with >26 disaccharide units represented approximately 10%–15% of the total glycan material (*Boneca et al., 2007*). CF-2's estimated molecular weight (24.93 kDa) along with its carbohydrate moiety content (50.70% ± 0.59%) yielded an average chain length of approximately 35 disaccharide units (Table 2).

Wild-type *Escherichia coli* strains in exponential growth have polysaccharides that are 25–35 disaccharide units long (*Pisabarro, De Pedro & Vazquez, 1985*; *Pisabarro et al.,*

*1987*; *Glauner, 1988*; *Glauner & Höltje, 1990*), and the glycan strands in the peptidoglycan of bacilli (including *Bacillus subtilis*, *B. licheniformis*, and *B. cereus*) have an average chain length of 50–250 disaccharide units (*Hughes, 1971*; *Warth & Strominger, 1971*; *Ward, 1973*). We believe that extremely long glycan strands might have relatively long non-crosslinked stretches; however, their presence is dependent on the relative chain number of the crosslinking peptides. However, biologically, large peptidoglycan content is essential to preserving the physical integrity, normal shape, and growth parameters of a cell (*Prats & De Pedro, 1989*; *Caparros, Pisabarro & De Pedro, 1992*). We thus propose that the long 35 disaccharide stretches with peptide crosslinking in its bran facilitates RQ in maintaining grain structure integrity, preventing pests and diseases, and more specifically, tolerating harsh drought climates.

### Absence of glutamic acid and hydroxyproline in RQ bran

Notably, all isolated polysaccharide fractions lacked glutamic acid and hydroxyproline (Table 4). Hydroxyproline occurs in higher plants as part of peptides and proteins found in their cell walls. A considerably larger amount of hydroxyproline is found in the proteins of rapidly proliferating tissues than in those of slowly proliferating tissues, mainly because it has structural and regulatory roles (*Olson, 1964*). Recent research has indicated that proline is crucial for growth and differentiation throughout a plant's lifecycle (*Kishor et al., 2015*). It is a key determinant of many cell wall proteins, which are essential for plant development (*Kishor et al., 2015*).

Drought and salinization impose a great threat to agriculture and food production worldwide. Therefore, understanding how plants defend themselves against damage related to low water levels or high osmolarity is essential. Many plant species accumulate proline or other compatible osmolytes in response to drought or salinity stress (*University of Konstanz, 2019*). However, in this study, we noted a simultaneous lack of glutamic acid and hydroxyproline, possibly implicating the lack of mitochondrial pyrroline-5-carboxylate dehydrogenase (P5CDH)—needed for conversion of ornithine to glutamate—as well as that of pyrroline-5-carboxylate synthase-1 (P5CS1) and pyrroline-5-carboxylate synthase-2 (P5CS2)—required for conversion of glutamic acid to proline in RQ bran cytosol. However, the reason that RQ tolerates very severe environments, including very low ($-4\,°C$) to high ($35\,°C$) temperatures and low rainfall (*U.S. National Research Council, 1989*), remains unclear.

### Strong antioxidative activity of CF-2

Of the four polysaccharide fractions from RQ, CF-2 showed the most potent FRSCs (Table 5). The order of magnitude was CF-2 > CF-1 > CF3 DPPH FRSC ($IC_{50} = 3.98$ mg/mL; Fig. 2A) and for hydroxyl FRSC ($IC_{50} = 4.10$ mg/mL; Fig. 2B), whereas for superoxide FRSC ($IC_{50}$ = slightly below 0.05 mg/mL), it was CF-2 > > CF-3 > CF-1, far stronger than the reference Trolox (Fig. 2C). However, all the polysaccharide fractions demonstrated extremely poor $Fe^{2+}$ chelation capability (Fig. 2D).

CF-2 was found to be far superior to fucoidans isolated from wakame and nori seaweeds—both of which showed high superoxide FRSC ($IC_{50} = 2.29 \pm 0.61$ mg/mL

for wakame and 2.80 ± 0.33 mg/mL for nori alga) (*Dimitrova-Shumkovska, Krstanoski & Veenman, 2020*).

Consistent with our findings, Yao et al. reported that both the water- and alkali-extracted polysaccharides demonstrate strong antioxidant and immunoregulatory activities (*Yao, Shi & Ren, 2014*).

## CONCLUSION

RQ demonstrates several differences compared with common cereals such as wheat, corn, rice, and oats; in particular, it is easy to grow and requires relatively less water. It is also nutritionally superior to many other cereals. RQ bran exhibits extremely high fibers, ash, phenolic contents (including betaxanthin and indicaxanthin) as well as nutritionally favorable fatty acid patterns (PUFA:SFA = 0.743, n-6/n-3 = 18.137). RQ is also gluten-free; therefore, it can be used in the development of gluten-free formulated foods. Moreover, RQ bran, which is considered an industrial or agricultural waste product, can have value, considering the diversity of high-cost nutritionally beneficial compounds present in it, including leucinoglucan (*i.e.,* CF-2), betalains, and polyphenolics. Here, CF-2 was noted to be a highly efficient superoxide scavenger.

Taken together, our results confirmed that not only RQ but also its bran has great nutritional value. However, for effective utilization of RQ bran, a cost-effective process is needed. Moreover, further research on the nutraceutical and biomedical applications of RQ bran is warranted.

## ACKNOWLEDGEMENTS

We thank Mr. Po-Wei Wang with San Gin Starches Co., Ltd. (Tainan City, Taiwan) for supplying all samples required for experiments and supporting many relevant instruments and technological guidance to carry out this research.

### Funding
The authors received funding from the The Ministry of Science and Technology (MOST 110-2314-B-038-068-) and Taipei Medical University-Shuang Ho Hospital (grant nos. 109TMUSHH-12 and 110TMU-SHH-13). The funders had no role in study design, data collection and analysis, decision to publish, or preparation of the manuscript.

### Grant Disclosures
The following grant information was disclosed by the authors:
The Ministry of Science and Technology: MOST 110-2314-B-038-068-.
Taipei Medical University-Shuang Ho Hospital: 109TMUSHH-12, 110TMU-SHH-13.

### Competing Interests
The authors declare there are no competing interests.

## Author Contributions

- Yaw-Bee Ker performed the experiments, analyzed the data, authored or reviewed drafts of the paper, and approved the final draft.
- Hui-Ling Wu performed the experiments, prepared figures and/or tables, and approved the final draft.
- Kuan-Chou Chen analyzed the data, prepared figures and/or tables, and approved the final draft.
- Robert Y. Peng conceived and designed the experiments, analyzed the data, authored or reviewed drafts of the paper, and approved the final draft.

## Data Availability

The raw data is available in the Supplemental Files.

## Supplemental Information

Supplemental information for this article can be found online at http://dx.doi.org/10.7717/peerj.13459#supplemental-information.

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
