# Peer review of "Nutrient composition of Chenopodium formosanum Koidz. bran: Fractionation and bioactivity of its soluble active polysaccharides"

_PeerJ, doi:10.7717/peerj.13459_

## Round 0.1 · original submission · Major Revisions

Please address the reviewers' comments very carefully. Your revised manuscript may be subjected to further peer review.

Reviewer 2 has suggested that you cite specific references. You are welcome to add it/them if you believe they are relevant. However, you are not required to include these citations, and if you do not include them, this will not influence my decision.

Reviewer 1 ·

Basic reporting

This MS aimed to design a laboratory systemic extraction process to isolate the polysaccharides from the bran of djulis. However,
1.There are nine figures and nine tables. This MS should concise and reorganize to make it more clear and readable. Many data are redundant and not necessary.
2.The mechnism or principle of seperation process should be mentioned in the introduction and compare with referrence in the discussion. In the conclusion, the contribution of this new process should be highlighted, not the plant itself.
3.Djulis is not quinoa. Red Quinoa is the name of one kind of the quinoa. In F9, the material seems not true for djulis grain. Furthermore, since indicaxanthin belongs to betaxanthain, and exhibits yellow color, the author should explain why measuring indicaxanthan at 532 nm (red color)?
4.It seems that the author is not familiar with the djulis material, I suggest the author check thoroughly and resubmit after rewriting.

Experimental design

The experiment seems not been well designed. The relevant and meaning of the research is not clear.

Validity of the findings

1.For a new extraction process, the contribution of this MS is not clearly stated.
2.Most part of the conclusion should be shift to the part of introduction.

Reviewer 2 ·

Basic reporting

1.This manuscript was well-written, experiment design and quality were enough for the requirement of this journal.
2. The literature's year author used in this introduction was too old, the author should add recent research related to Chenopodium formosanum. https://doi.org/10.3390/ma14164679; https://doi.org/10.3390/biology10020160

Experimental design

The author should show the growth year of this plant.

Validity of the findings

1.The number of equation should be added.

Reviewer 3 ·

Basic reporting

The manuscript is fairly easy to read and understand but contains some grammatical errors and incomplete sentences. The manuscript should be proofread by a native English speaker to increase clarity in many sections. The manuscript, in my opinion, is too long due to excessive description of textbook facts and repetition of results in the discussion section.
The phrase "Development of a Systemic Extraction Process for the Unique Polysaccharide" in the title should be removed as this is only part of the fractionation process for polysaccharides; the elements of optimisation are absent, hence, in my opinion, it is inaccurate to say "development of systemic xxx".
The conclusion should be short and concise - unnecessary factual accounts should be omitted.

Experimental design

The polysaccharide fractions have to be heated for dissolving them in the water, sonicated, heated overnight(?) and it was mentioned that only the supernatant was used (lines 284-286).
Which apparatus was used for overnight heating at 150 degrees Celsius (precise measurement of temperature)?
Provide the breakdown of the water-soluble and water insoluble fractions of the polysaccharides.
If only the supernatants were used, the authors actually accessed only the water-soluble fractions.
It is uncommon to come across procedures where extracts/fractions have to be heated overnight prior to analysis. I am concerned about potential changes to chemical composition or structures of compounds in the fraction as a result of this prolonged heating. It could be incompatible solvent (phosphate buffer) being used. Did the authors try other solvents to dissolve the fractions and/or to test at lower concentrations?
The choice of positive controls used in the antioxidant assays needs to be justified, for instance, why trolox was not used in place of BHA for the DPPH assay. IC50 values of the fractions in the antioxidant assays should have been included.
The MTT assay is commonly used to evaluate the effect of test substances on cell viability. It does not provide any information on the potential antiinflammatory activity of the polysaccharides as the authors claimed (lines 655-656). There are other established assays for assessment of antiinflammatory activity.
Statistical analysis is not clear or absent. In Figures 6 & 8, error bars are missing.
Justify the use of only fraction CF2 for cell viability analysis.
How did the authors determine the IC50 values?

Validity of the findings

The polysaccharides demonstrated antioxidant activity but at very high concentrations (in the range of mg/ml), hence, claims about the "potency" of the extracts should be revised. These values should be compared against to those in the literature. What is the authors' definition of "potent" samples?
There are many unsubstantiated claims (e.g., antiinflammatory activity) throughout the manuscript, for instance, "peptidoglucan CF-2, showing strong antioxidative and anti inflammatory bioactivity, speculatively, could become a rather useful medicine" (lines 669-670) and " not only exhibits potent antioxidative capability, but also shows a promising anti-inflammatory bioactivity in the RAW264.7 cell model" (lines 698-699).

Additional comments

The manuscript reports on the chemical characterisation and bioactivities evaluation of red quinoa. In particular, the polysaccharides were systematically fractionated and the resulting fractions were subjected to chemical analysis, antioxidant and cell viability assays. The authors are commended for the large amount of data collected, however, analysis and interpretation of the results should be done in a careful manner. Specifically, avoid making unsubstantiated claims and erroneous conclusions (e.g., the claims about antiinflammatory activity based on MTT assay). In the discussion section, findings from this study should be compared to the literature on the quinoa but this is lacking. The proposed design for pilot scale process has nothing to do with the bulk data presented.

---

## Round 0.2 · Major Revisions

I am concerned about the points raised by Reviewer 3. The authors are requested to consider these comments and provide convincing responses to them.

Reviewer 2 ·

Basic reporting

The author had corrected the manuscript according to my comments.

Experimental design

The author had corrected the manuscript according to my comments.

Validity of the findings

The author had corrected the manuscript according to my comments.

Additional comments

The author had corrected the manuscript according to my comments.

Reviewer 3 ·

Basic reporting

The new title (i.e., inclusion of the term 'breakdown') is misleading and does not reflect the content of the manuscript.
Data presentation and analysis needs improvement. There are simply too many figures and tables in this manuscript.
Many of these duplicate the same data set, for instance Fig. 3 with Table 4, Fig. 4 with Table 5 and Fig. 5 with Tables 6-9.
The chromatograms can be included as supplementary information instead. The same goes for the dose-response curves of the antioxidant assays, which are redundant if the outcome has been summarised in the form of IC50 values.
Some are unnecessary, such as Table 1. The the information can be included in the text itself.
The legend for Fig. 5A does not tally with the graph.
Objective writing is crucial in scientific manuscript. Subjective and emotive terms, for instance, in line 443 (astonishment) and 544 (amazingly), might be unsuitable.

Experimental design

The inclusion of a prolonged heating step in extracts preparation (i.e.50 or 150 degrees Celsius depending on the antioxidant assay) needs to be explained. The authors should try to use water or to prepare stock at lower concentrations. My concern would on the possibility of degradation of the more sensitive heat sensitive chemical components in the extracts - which has not been properly studied. If a centrifugation step is still required after that prolonged heating and only the supernatant was used for the assays (suggesting that there is a significant amount of precipitate), then the overnight heating, in my opinion, cannot be justified. This is a major flaw of the present work.

Validity of the findings

The authors added the word "breakdown" in the title but there wasn't any work done to determine if there was indeed any decomposition of the polysaccharides and/or other chemical components in the extracts.

In fact, there was very little attempt to correlate the antioxidant activity with the chemical composition of the samples and/or polysaccharides in the entire manuscript.

The word "breakdown" in my previous review were misunderstood. I was referring the supernatant and precipitate fractions that were produced during the extracts preparation step.

It was claimed that the polysaccharides were good antioxidants (lines 443), particularly CF2 as "efficient" superoxide scavenger (line 623) but there was no other polysaccharides/extracts/samples that were included in the study for comparison purposes. How did the authors conclude that IC50 values of 0.26, 0.05, and 0.19 mg/mL can be considered as good/active?

The results could have been compared with the extensive literature on antioxidant capacity of polysaccharides/extracts so that a more rational conclusion can be drawn from this study.

Additional comments

The authors have addressed only some of the comments raised by the reviewers satisfactorily and modified the manuscript but there are many other issues that need to be addressed.

---

## Round 0.3 · Major Revisions

Following the comment of the Section Editor, the quality of writing is not acceptable for publication. The authors should seek help from a professional proofreading service and provide a certificate. Especially errors in grammar and word choice make the text difficult to understand. An example from the Conclusions: "Taken together, the growing of RQ is worth being inspired." (What do the authors imply). The tenses (past and presence) are often wrong.

In methods, the authors should report centrifugation parameters in g-force, not in RPM.

---

## Round 0.4 · accepted · Accept

The manuscript has been edited, and it has improved considerably.